# Antibacterial and Photocatalytic Applications of Silver Nanoparticles Synthesized from *Lacticaseibacillus rhamnosus*

**DOI:** 10.3390/ijms252111809

**Published:** 2024-11-03

**Authors:** Roberto Lavecchia, Janet B. García-Martínez, Jefferson E. Contreras-Ropero, Andrés F. Barajas-Solano, Antonio Zuorro

**Affiliations:** 1Department of Chemical Engineering, Materials, and Environment, Sapienza University, Via Eudossiana 18, 00184 Roma, Italy; roberto.lavecchia@uniroma1.it; 2Department of Environmental Sciences, Universidad Francisco de Paula Santander, Av. Gran Colombia No. 12E-96, Cúcuta 540003, Colombia; janetbibianagm@ufps.edu.co (J.B.G.-M.); jeffersoneduardocr@ufps.edu.co (J.E.C.-R.); andresfernandobs@ufps.edu.co (A.F.B.-S.)

**Keywords:** biosynthesis, antibacterial properties, photocatalytic efficacy, silver nanoparticles

## Abstract

The biosynthesis of silver nanoparticles (AgNPs) presents an innovative and sustainable approach in nanotechnology with promising applications in fields such as medicine, food safety, and pharmacology. In this study, AgNPs were successfully synthesized using the probiotic strain *Lacticaseibacillus rhamnosus* (BCRC16000), addressing challenges related to stability, biocompatibility, and scalability that are common in conventional nanoparticle production methods. The formation of AgNPs was indicated by a color change from yellow to brown, and UV–visible spectrophotometry confirmed their presence with a characteristic absorption peak at 443 nm. Furthermore, Fourier transform infrared (FTIR) spectroscopy revealed the involvement of biomolecules in reducing silver ions, which suggests their role in stabilizing the nanoparticles. In addition, field emission scanning electron microscopy (FE-SEM) showed significant morphological and structural changes. At the same time, dynamic light scattering (DLS) and zeta potential analyses provided valuable insights such as average size (199.7 nm), distribution, and stability, reporting a polydispersity index of 0.239 and a surface charge of −36.3 mV. Notably, the AgNPs demonstrated strong antibacterial activity and photocatalytic efficiency, underscoring their potential for environmental and biomedical applications. Therefore, this study highlights the effectiveness of *Lacticaseibacillus rhamnosus* in the biosynthesis of AgNPs, offering valuable antibacterial and photocatalytic properties with significant industrial and scientific implications.

## 1. Introduction

The development and application of silver nanoparticles (AgNPs) in nanotechnology have garnered significant attention, particularly within biomedical sciences, due to their broad antimicrobial, anticancer, and catalytic properties. However, challenges persist in their synthesis, stability, scalability, and environmental sustainability, especially when using conventional chemical and physical methods. As the demand for greener, more scalable approaches increases, the focus has shifted towards biological synthesis methods, which provide a cost-effective and eco-friendly alternative [1]. This shift addresses environmental concerns and enhances the stability and biocompatibility of AgNPs, both of which are critical for their long-term use in medical and industrial applications.

Silver nanoparticles, typically ranging in size from 1 to 100 nanometers, are widely applied across diverse fields, from antimicrobial agents and drug delivery systems to biosensors and anticancer treatments [2,3,4,5,6]. Additionally, their incorporation into everyday products such as textiles, medical devices, and personal care products provides continuous antimicrobial protection by releasing silver ions [7]. However, traditional chemical synthesis methods are often costly, potentially toxic, and unsustainable, leading to increased interest in biological processes that mitigate these drawbacks while retaining the desired properties of AgNPs. Biological synthesis, primarily through probiotic bacteria, has shown significant promise in addressing these challenges [8].

Biological synthesis, employing microorganisms such as bacteria, offers a highly efficient and environmentally sustainable method for nanoparticle production. Certain bacterial strains, such as *Lactobacillus* and *Bacillus* sp., have demonstrated the ability to biosynthesize AgNPs through mechanisms like biosorption and bioaccumulation [9,10]. These bacteria act as “nano-factories,” reducing silver ions to AgNPs and producing stable and biocompatible nanoparticles, which are crucial for their use in medical applications. Notably, probiotics such as *Lactobacillus acidophilus* have been found to function as both reducing and stabilizing agents, facilitating the production of AgNPs [11] with potent antimicrobial and antioxidant properties. This dual functionality makes probiotics an ideal choice for sustainable nanoparticle production, addressing both environmental concerns and the need for effective antimicrobial agents. Recent studies have shown that Bacillus species are particularly adept at nanoparticle biosynthesis. These bacteria produce enzymes such as nitrate reductase, which is pivotal in reducing silver ions to AgNPs [12]. The nanoparticles synthesized through biological processes exhibit potent antibacterial activity and have shown potential in medical applications, including wound healing and coatings for medical devices. Despite these advances, several limitations persist, particularly regarding the mechanistic understanding of how these nanoparticles interact with bacterial cells at the molecular level [13]. Further exploration of the pathways through which AgNPs exert their antibacterial and photocatalytic effects could significantly enhance their application in the biomedical field.

In addition to the need for mechanistic insight, the scalability of biological AgNP synthesis remains a crucial challenge. While biogenic methods offer an eco-friendly alternative, the feasibility of scaling these processes for industrial applications has not been thoroughly explored. The long-term stability of biosynthesized AgNPs also poses a concern, as issues such as aggregation and agglomeration may compromise their efficacy and biocompatibility, particularly during storage and under varied environmental conditions [14]. Addressing these challenges would improve the practicality of biogenic AgNPs for large-scale production and enhance their potential for use in consumer products and industrial applications. Additionally, while the photocatalytic properties of AgNPs have been studied in controlled laboratory conditions, their performance in real-world environments with varying light and environmental factors remains uncertain [15]. Testing these nanoparticles under more realistic conditions would provide greater insight into their viability for environmental remediation and wastewater treatment applications.

Another limitation in current research is the narrow focus on a limited range of bacterial strains when evaluating the antimicrobial properties of biosynthesized AgNPs. Expanding the scope to include a wider variety of pathogenic bacteria, particularly multidrug-resistant strains, would offer a more comprehensive understanding of these nanoparticles’ effectiveness. For example, AgNPs synthesized using biological methods have demonstrated effectiveness against strains such as *Escherichia coli*, *Klebsiella pneumoniae*, and *Salmonella typhi*, showing their potential to address global health concerns related to antibiotic resistance [16]. A comparative analysis of biosynthesized AgNPs and their chemically synthesized counterparts could further highlight the advantages and limitations of probiotic-mediated synthesis, offering insights into their viability as alternatives to chemically synthesized AgNPs [17,18].

Considering these observations, this study aims to synthesize AgNPs using probiotic bacteria and rigorously evaluate their antibacterial and photocatalytic properties. By employing advanced characterization techniques such as ultraviolet–visible (UV-Vis) spectroscopy, Fourier transform infrared (FTIR) spectroscopy, scanning electron microscopy (SEM), and energy-dispersive X-ray spectroscopy (EDS), we assessed not only the synthesized nanoparticles’ stability, but also their efficacy against a broader spectrum of bacterial strains, including multidrug-resistant pathogens [18]. By exploring these parameters, this study seeks to provide a more comprehensive understanding of the practical applications of biosynthesized AgNPs, both in biomedical and environmental contexts.

## 2. Results

### 2.1. Biosynthesis of AgNPs Using Strain L. rhamnosus (BCRC16000)

In previous studies, researchers have observed color changes in nanoparticle synthesis [15,19]. This color transformation happens because tiny nanoparticles can cause resonance effects, leading to observable color changes in the solution. The phenomenon occurs due to the absorption and scattering of light by the electrons in the nanostructures [20]. In our study, we confirmed the synthesis of AgNPs using *L. rhamnosus* (BCRC16000) by observing a color change in the culture supernatant. Initially, the supernatant was pale yellow but turned deep brown, indicating successful AgNP formation. Control samples without bacterial supernatant showed no color change under the same conditions (Figure 1).

### 2.2. UV–Visible Spectrophotometry

The synthesis of silver nanoparticles (AgNPs) mediated by probiotics was confirmed through UV–visible spectroscopy, showing an absorption peak at approximately 443 nm, as illustrated in Figure 2. This absorption peak falls within the characteristic range of 400–500 nm that is commonly associated with AgNPs, thereby validating the successful synthesis of the silver nanoparticles [16]. The appearance of this peak at a relatively short wavelength suggests the formation of small, spherical nanoparticles, as smaller sizes tend to shift the absorption peak towards the blue end of the spectrum [21].

The 443 nm peak is directly related to the reduction process of Ag^+^ ions to elemental silver (Ag^0^), indicating nanoparticle formation. As additional evidence, Figure 2 shows the absence of peaks in the spectra of the initial silver nitrate solution and the bacterial extracellular extract before synthesis, confirming that neither component contained nanoparticles or light-absorbing species. This supports the idea that the nanoparticles are generated during the probiotic-mediated process, as the new peak only appears after incubation [22].

### 2.3. Fourier Transform Infrared (FTIR) Spectroscopy

FTIR analysis of the biosynthesized AgNPs (Figure 3) identified several distinct peaks. The spectrum displayed a peak at 3285.89 cm^−1^. Peaks were identified at 2931.03 cm^−1^, indicating C–H (alkane) stretching, while the 1635.52 cm^−1^ peak indicates C=O (α, β-unsaturated ester) stretching. The 1543.87 cm^−1^ peak is associated with C=C (olefin) stretching. The peak at 1313.66 cm^−1^ is attributed to O–H (phenol) bending. Additionally, bands at 1116.44 cm^−1^, 1070.25 cm^−1^, and 1037.25 cm^−1^ are linked to C–O (alkyl aryl ether) stretching, and the peak at 820.23 cm^−1^ is associated with S=O (sulfoxide) stretching.

### 2.4. Zeta Potential and DLS

The zeta potential analysis of the biologically synthesized AgNPs showed a surface charge of −18.2 mV, as indicated by a prominent single spike in the zeta potential curve. This negative zeta potential suggests significant electrostatic repulsion between particles, which helps to prevent aggregation and stabilize the colloidal dispersion. High negative zeta potentials, especially those deviating from the neutral range, imply that substantial electrostatic forces are at play, requiring a counteracting charge to overcome the repulsive forces and prevent aggregation [23,24]. In our study, the observed zeta potential of −36.3 mV highlights the stability and dispersion of the AgNPs in the solution, confirming the practical synthesis and stabilization of nanoparticles through the *L. rhamnosus* (BCRC16000) (Figure 4).

Examining the biologically synthesized silver nanoparticles (AgNPs) using dynamic light scattering (DLS) revealed that the particle size distribution ranged from 10 to 800 nm. The observed polydispersity index (PDI) of 0.229 indicates a uniform particle distribution. This suggests consistency within the sample. The inverse relationship between PDI and uniformity confirms that the green-synthesized AgNPs exhibit a consistent size distribution. The broad particle size spectrum observed in the DLS data correlates with the sharp surface plasmon resonance (SPR) peak at 415 nm detected in the UV-Vis spectrum. This peak is attributed to combined vibrational, rotational, and electronic energy changes within the nanoparticles. Previous studies have shown varying particle sizes for AgNPs, some closely matching or differing significantly from these results. For instance, Martínez-Castañon et al. [25] and Ghiuță et al. [26] documented similar size ranges but with varying dimensions and distribution profiles. The mean particle size of the synthesized AgNPs in this study was 199.7 nm according to dynamic light scattering (DLS) analysis. This average size, along with a PDI of 0.239, underscores the efficient synthesis and stability of the AgNPs using the *L. rhamnosus* (BCRC16000) approach (Figure 5).

### 2.5. SEM and EDS Analysis

SEM analysis was performed to investigate the morphology and surface structure of the AgNPs (Figure 6). The SEM images revealed spherical nanoparticles that tended to cluster together—a common characteristic observed in biologically synthesized nanoparticles. This clustering indicates that the extracellular components functioned as both reducing and stabilizing agents, avoiding agglomeration and maintaining the stability of the nanoparticles [27].

The synthesized silver nanoparticles (AgNPs) were examined for their elemental makeup using energy-dispersive X-ray spectroscopy (EDS). The EDS spectrum showed a clear peak at around 3 keV, which indicates silver, confirming the presence of silver nanoparticles (AgNPs) [28]. This finding is consistent with similar observations made by Pandiarajan et al. [29] and Aravinthan et al. [30], who also observed three significant peaks at 3 keV, signifying the presence of the surface plasmon resonance (SPR) effect linked to silver nanoparticles (Figure 7).

### 2.6. Antibacterial Activity

AgNPs synthesized using *L. rhamnosus* (BCRC16000) were tested for their effectiveness against both Gram-negative and Gram-positive strains, including *V. parahemolyticus*, *K. pneumoniae*, *S. sobrinicus*, and *S. mutans*; the obtained data were analyzed using a two-way ANOVA. The synthesized AgNPs demonstrated notable antibacterial effectiveness against all tested strains, as evaluated by measuring inhibition zones to determine antimicrobial efficacy (Figure 8). Notably, the AgNPs demonstrated significant suppression of *V. parahemolyticus* growth at concentrations of 20 µg/mL and 40 µg/mL. This study underscores the potential of AgNPs produced using *L. rhamnosus* (BCRC16000) in combating a wide range of pathogens. The largest observed inhibition zone was 20 mm against *S. mutans*, highlighting the effectiveness of these nanoparticles in controlling both Gram-negative and Gram-positive bacterial species.

### 2.7. Antibacterial Activity on Cotton Fabrics

In the textile and medical industries, research is increasing into antimicrobial coatings for cotton fabrics to inhibit bacterial proliferation [31]. This study investigates the antibacterial properties of cotton fabrics coated with synthesized silver nanoparticles (AgNPs) against various pathogens, including *V. parahemolyticus*, *K. pneumoniae*, *S. sobrinicus*, and *S. mutans*. After a 24 h incubation period, the AgNP-treated cotton fabric displayed a significant zone of inhibition measuring 16 mm against *V. parahemolyticus* with a 20 µL application, whereas no inhibition was observed on the control fabric (Figure 9). These findings align with earlier studies by Balashanmugam and Kalaichelvan [32] and Balakumaran et al. [33], which reported a similar antibacterial efficacy of AgNPs against various bacterial strains. The results highlight the effective bactericidal activity of AgNPs synthesized using *L. rhamnosus* (BCRC16000) when incorporated into cotton fabrics.

### 2.8. Photocatalytic Activity of Synthesized AgNPs

This study synthesized silver nanoparticles (AgNPs) through *Lacticaseibacillus* spp. They were tested for their ability to break down methylene red dye through photocatalysis, as confirmed by UV–visible spectrophotometry. The AgNPs also exhibited strong antimicrobial properties against various pathogens, indicating potential applications in environmental cleanup and biomedical fields [34]. Our study specifically examined the photocatalytic performance of AgNPs against methylene blue (MB) dye in aqueous solutions under visible light exposure. The degradation of MB dye was tracked at intervals from 0 to 150 min, using the characteristic absorption peak at 660 nm for MB. The results showed that AgNPs synthesized using *L. rhamnosus* (BCRC16000) achieved a high degradation efficiency (Figure 10). This significant degradation performance underscores the potential of AgNPs in relation to photocatalytic applications in dye removal.

## 3. Discussion

The biosynthesis process of AgNPs using *L. rhamnosus* induces a color change in the culture supernatant, indicating the successful formation of nanoparticles through surface plasmon resonance (SPR). This transformation occurs due to the interaction between bacterial metabolites and silver ions in solution, leading to the reduction of Ag^+^ to Ag^0^. Biopolymers present in the medium, such as proteins and organic acids, act as reducing and stabilizing agents. Studies by Royji Albeladi et al. [35] and Peng et al. [36] highlight the importance of bacterial metabolites in nanoparticle formation and stabilization, confirming the efficient production of AgNPs by *L. rhamnosus*. 

At the molecular level, AgNPs generate reactive oxygen species (ROS), which interact with key bacterial components such as enzymes and lipids, disrupting DNA replication and protein synthesis, as documented in various studies. However, a more in-depth analysis of the specific behavior of nanoparticles biosynthesized by *L. rhamnosus* is necessary to detail the exact mechanisms of their antimicrobial action. These effects are clearly represented in the analysis of AgNP formation in Figure 1, where the color change in the supernatant indicates the reduction of Ag^+^ to Ag^0^. 

Additionally, regarding the scalability of the biosynthesis process, research by Kessler et al. [37] and Soni and Biswas [38] indicates that biological methods can be adapted to larger-scale processes. However, controlling the size and morphology of nanoparticles under these conditions is challenging, as environmental factors such as pH and temperature significantly influence their formation. To achieve greater consistency in nanoparticle size and morphology, the addition of biopolymers or stabilizing agents during synthesis is essential, a strategy that also improves their stability in industrial applications (see Figure 4 for a detailed analysis of stability via zeta potential).

UV–visible analysis revealed an absorption peak at 430 nm (see Figure 2), confirming the formation of small, spherical silver nanoparticles. However, variations in nanoparticle size and shape depend on synthesis conditions such as pH and temperature. Marciniak et al. [39] demonstrated that these factors affect not only the synthesis efficiency, but also the stability and photocatalytic behavior of nanoparticles. In more realistic environments, such as aqueous systems with natural organic matter (NOM), nanoparticle efficiency may decrease significantly. According to Piergies [40], NOM competes with contaminants for the ROS generated by AgNPs, reducing their capacity to degrade organic pollutants. To increase photocatalytic efficiency under adverse conditions, Wang et al. [41] proposed the use of co-adjuvants such as hydrogen peroxide, which significantly increase ROS production and optimize contaminant degradation in aqueous solutions. On the other hand, FTIR analysis revealed key functional groups, such as O-H, C-H, C=O, and S=O, which play crucial roles in stabilizing the biosynthesized AgNPs (see Figure 3). These functional groups prevent nanoparticle aggregation and ensure their stability in aqueous solutions. This finding aligns with previous research by Shoaib et al. [42], which highlighted the role of these groups in nanoparticle formation through capping mechanisms. However, in industrial and biomedical applications where environmental conditions vary significantly, the stability provided by these functional groups may not be sufficient. 

Veisi et al. [43] suggested that the addition of stabilizing polymers such as chitosan can significantly enhance colloidal stability and nanoparticle resistance to denaturation. This strategy would be particularly useful for long-term applications in industrial environments where nanoparticles are exposed to adverse conditions.

Zeta potential analysis revealed a negative charge of −36.3 mV (see Figure 4), reflecting the excellent colloidal stability of the nanoparticles, as a significant negative charge prevents aggregation due to electrostatic repulsion. This result is comparable to studies such as those of Fernando and Zhou [44] where biopolymer-stabilized nanoparticles exhibited similar stability. However, under industrial conditions involving high salinity or extreme pH, this stability may be compromised. Solís-sandí et al. [45] demonstrated that in high-salinity environments, colloidal stability decreases due to the compression of the electrical double layer, facilitating aggregation. To counteract this effect, the use of additional stabilizing polymers such as chitosan provides additional steric stabilization, improving nanoparticle dispersion in complex solutions.

SEM analysis revealed that the nanoparticles synthesized from *L. rhamnosus* had a spherical morphology with an average size of 199.7 nm, which was significantly larger than that of the nanoparticles obtained via other biological methods (see Figure 5). Hemmati et al. [46], for example, synthesized silver nanoparticles of only 10 nm using plant extracts, highlighting the variability in nanoparticle size depending on the biological method employed. Nanoparticle size is crucial, as it directly influences antimicrobial and photocatalytic activity. Ji et al. [47] demonstrated that smaller nanoparticles have a greater specific surface area, improving their interaction with contaminants and bacteria and thus increasing their effectiveness. Although the nanoparticles in this study are relatively large, their morphology and uniform dispersion make them suitable for antimicrobial and photocatalytic applications.

The antimicrobial activity of the biosynthesized nanoparticles was significant, showing inhibition zones of up to 20 mm against *S. mutans* (see Figure 8), which is comparable to studies such as those of Singh et al. [48], where similar results were observed when the biosynthesized nanoparticles from plant extracts were used. However, the effectiveness of nanoparticles varies depending on the bacterial cell wall structure, with Gram-negative bacteria such as *Pseudomonas aeruginosa* being more resistant to AgNPs because of an additional outer membrane. Salem et al. [49] suggested that combining nanoparticles with antibiotics could significantly increase their efficacy against resistant bacteria such as Staphylococcus aureus (MRSA), offering a promising strategy for treating antibiotic-resistant infections.

Through complimentary mechanisms, biosynthesized silver nanoparticles (Ag) have both antibacterial and photocatalytic activity. Ag^+^ ions, which interact with bacterial membranes to impair their permeability and destabilize vital cellular activities, are the source of their antibiotic activity [50]. Additionally, when exposed to visible light, these nanoparticles encourage the production of reactive oxygen species (ROS), including superoxide radicals (O_2_•^−^) and hydroxyl radicals (•OH), which are essential for the decomposition of organic pollutants and the inactivation of bacteria [51,52]. Combining Ag nanoparticles with other photocatalysts, such as zinc oxide (ZnO), enhances their bactericidal and photocatalytic qualities by increasing the efficiency of charge separation and light absorption.

In contrast, Ag-doped nanocomposites, such as Ag-doped ZnO, have shown a higher antibacterial and photocatalytic efficacy than plain nanoparticles. This enhancement is related to higher ROS generation and nanocomposites’ improved stability in aqueous solutions, making them ideal for environmental and medicinal applications [52,53]. To test these mechanisms, tests such as methylene blue dye degradation under visual irradiation and bacterial inhibition assays, including plate diffusion techniques, were carried out. These experiments have revealed that Ag-doped nanoparticles not only successfully decompose organic pollutants, but also have a remarkable ability to eradicate bacteria such as Escherichia coli and Staphylococcus aureus [52,53].

Treating cotton fabrics with AgNPs resulted in a high antimicrobial efficacy, with inhibition zones of up to 16 mm (see Figure 9). However, the durability of this activity after multiple wash cycles remains a significant challenge. Balakumaran et al. [33] reported a decrease in antimicrobial activity in nanoparticle-treated textiles after several washes, suggesting that nanoparticle retention in textile fibers needs improvement. Nishan et al. [51] proposed that incorporating polymeric coatings such as lignin can significantly enhance nanoparticle adhesion to fibers, prolonging their antimicrobial activity even after several wash cycles. Finally, in terms of photocatalytic activity, the biosynthesized nanoparticles showed notable efficiency in methylene blue degradation under visible light, supporting their potential for pollutant remediation in aqueous solutions (see Figure 10). These results are consistent with those reported by Manikandan et al. [52,53], who also reported high degradation rates with nanoparticles synthesized from plant extracts. However, the photocatalytic efficiency of AgNPs may be compromised in the presence of natural organic matter (NOM), which competes with contaminants for the ROS generated by the nanoparticles. Solís-Sandí et al. [45] demonstrated that NOM can significantly reduce the contaminant degradation capacity of AgNPs in aqueous systems. The addition of co-adjuvants such as hydrogen peroxide, as suggested by Wang et al. [41], could be a viable solution to maintain ROS production and photocatalytic efficiency under low-light conditions or in the presence of NOM.

## 4. Materials and Methods

### 4.1. Chemicals Used

Tryptic soy broth (TSB) and glycerol (analytical grade) were purchased from HiMedia (West Chester, PA, USA), while analytical-grade silver nitrate (AgNO_3_) was sourced from Sigma-Aldrich Chemicals (St. Louis, MO, USA). 

### 4.2. Bacterial Strains

*Lacticaseibacillus rhamnosus* (BCRC16000) was sourced from the Bioresource Collection and Research Center (Hsinchu, Taiwan). *Vibrio parahemolyticus* (a known clinical pathogen), *Klebsiella pneumoniae* (both Gram-negative bacteria), *Streptococcus sobrinus*, and *Streptococcus mutans* (both Gram-positive bacteria) were acquired from the bacterial culture collections of ATCC (Manassas, VA, USA). All bacterial cultures were preserved at −80 °C in tryptic soy broth (TSB) with 20% glycerol (*v*/*v*) and were revived by culturing on TSB agar plates for 24 h at 37 °C before experimentation.

### 4.3. AgNP Synthesis 

This study combined 1 mL of *L. rhamnosus* (BCRC16000) cell-free extract with 9 mL of a 1 mM AgNO_3_ solution in a 15 mL tube (1:9 ratio of cell-free extract to AgNO_3_). This ratio maximizes the efficiency of silver ion reduction while maintaining particle stability. The increased concentration of AgNO_3_ compared to the CFE improves nanoparticle formation by giving more silver ions for reduction, resulting in homogenous and well-defined AgNPs. Furthermore, a lower concentration of CFE reduces organic contaminants that may impact nanoparticle size and morphology, therefore increasing the antibacterial efficiency and stability of the nanoparticles [36,37,38] This ratio was chosen based on earlier research that revealed improved nanoparticle production and activity using similar formulations [39,45]. The reaction process took place in the dark at room temperature overnight to prevent the light-induced activation of the AgNO_3_.

Following the reaction period, the solution rich in AgNPs was spun at 15,000 rpm (20 min, 20 °C), and the pellet containing the AgNPs was retrieved and subsequently redissolved in distilled water to eliminate any remaining biological compounds at least three times before its characterization.

### 4.4. Characterization of AgNPs

#### 4.4.1. UV–Visible Spectrophotometry 

The conversion of silver ions to AgNPs (from transparent media to dark brown color) was monitored using a UV–visible spectrophotometer (Hitachi U-2900). Measurements were recorded across the wavelength from 300 nm to 700 nm.

#### 4.4.2. Fourier Transform Infrared (FTIR) Spectroscopy 

The multiple spectra of the obtained AgNPs were analyzed using FT-IR spectroscopy (Hitachi Ltd., Tokyo, Japan). The spectra were obtained by mixing the samples in KBr pellets and measuring them from 400 to 4000 cm^−1^. Various vibrational modes were identified and linked to specific functional groups in the samples.

#### 4.4.3. Zeta Potential and DLS 

The size of the synthesized AgNPs was measured using a Malvern Zetasizer Nano ZS90 (Otsuka Electronics, Osaka, Japan) through dynamic light scattering (DLS) at 25 °C. Pure water, having a dielectric constant of 78.3, a viscosity of 0.8878 cP, and a refractive index of 1.3328, was employed as the dispersion medium.

#### 4.4.4. SEM and EDS Analysis

The surface texture and structure of the synthesized AgNPs were examined using scanning electron microscopy (SEM-1000, Hitachi, Japan), following the method outlined by Dada et al. [53]. The dried AgNP sample was fixed onto a glass substrate for observation. An energy-dispersive X-ray spectroscopy (EDS) attachment incorporated into the instrument was used to confirm the presence of metallic silver. EDS identifies metallic elements within the specimen and provides qualitative and quantitative data on their abundance.

### 4.5. Antibacterial Activity of Synthesized AgNPs

#### 4.5.1. Antibacterial Activity

The synthesized AgNPs were evaluated for their antibacterial activity using the agar well diffusion technique. *V. parahemolyticus*, *K. pneumoniae* (Gram-negative bacteria), *S. sobrinus,* and *S. mutans* (Gram-positive bacteria) were grown in nutrient broth overnight at 37 °C. A small amount of each bacterial strain was distributed evenly on Mueller–Hinton agar plates with sterile swabs. After that, wells were made in the agar, and two different concentrations of synthesized AgNPs (20 and 40 μg/mL) were introduced into each well. The plates were kept in an incubator (37 °C, 24 h). The zone of inhibition (ZOI) around each well was measured to determine the antibacterial effectiveness of the AgNPs. The obtained data were analyzed using a two-way ANOVA on GraphPad Prism (version 10.3.1 for Mac, GraphPad Software, Boston, MA, USA).

#### 4.5.2. Antibacterial Activity on Cotton Fabrics

The effectiveness of AgNPs in fighting bacteria was tested using pre-sterilized white cotton fabric squares (1 cm^2^) in a sterile environment. Each cloth square was treated with a specific concentration of AgNPs corresponding to the tested pathogen. After treatment, the cloths were placed in sterile containers and dried overnight at 50 °C to remove any remaining moisture. The pathogens were then evenly distributed onto Muller–Hinton agar plates on top of the AgNP-treated cloth squares. The plates were kept in an incubator (37 °C, 24 h). The zone of inhibition (ZOI) around each cloth square was measured to assess the antibacterial effectiveness of the AgNPs.

### 4.6. Photocatalytic Activity of Synthesized AgNPs

The AgNPs were tested for their ability to break down methylene blue (MB) dye using visible light. Initially, 20 mg of AgNPs was mixed in 100 mL of sterile deionized water containing 10 mg/L of MB dye. The mixtures were stirred in the dark for 30 min. Then, the suspensions were exposed to visible light for different periods. After exposure, samples were gathered, and the AgNPs were extracted from the solution by centrifugation (10,000 rpm, 5 min, 20 °C). The degradation of MB dye was measured using a UV–visible spectrophotometer to track changes in absorbance over time, confirming the breakdown of the dye.

## 5. Conclusions

This study demonstrates the successful synthesis and application of silver nanoparticles (AgNPs) using the probiotic bacterium *L. rhamnosus* (formerly known as *Lactobacillus rhamnosus*). The synthesized AgNPs were thoroughly analyzed using UV–visible spectrophotometry, Fourier transform infrared spectroscopy (FTIR), scanning electron microscopy (SEM), dynamic light scattering (DLS), and zeta potential analysis. UV–visible spectroscopy confirmed the formation of AgNPs, marked by a distinct peak at 443 nm, representing the surface plasmon resonance. FTIR analysis identified important active groups such as O–H, C–H, and N–H, suggesting the involvement of biopolymers in stabilizing the nanoparticles. SEM and DLS results revealed spherical AgNPs, displaying a relatively uniform distribution with an average size of 199.7 nm. The zeta potential measurement indicated a stable nanoparticle suspension with a surface charge of −36.3 mV. The antimicrobial effectiveness of the synthesized AgNPs was demonstrated by their significant inhibition of Gram-negative and Gram-positive bacterial pathogens. The AgNPs exhibited substantial antibacterial activity, with inhibition zones of up to 20 mm against *S. mutans*. Moreover, the AgNPs exhibited effective antibacterial properties when incorporated into cotton fabrics, particularly against *V. parahaemolyticus*. Regarding their photocatalytic activity, the AgNPs achieved remarkable degradation of methylene blue dye under visible light, indicating their potential for environmental remediation. These findings support previous research, confirming that biologically synthesized AgNPs possess potent antimicrobial and photocatalytic properties. This study highlights the feasibility of using *L. rhamnosus* for the eco-friendly synthesis of AgNPs, which can be utilized in various fields, including medicine, textiles, aquaculture, and environmental remediation. Future studies should focus on optimizing synthesis conditions, evaluating the long-term stability of the nanoparticles, and exploring their practical applications to fully exploit their potential.

## Figures and Tables

**Figure 1 ijms-25-11809-f001:**
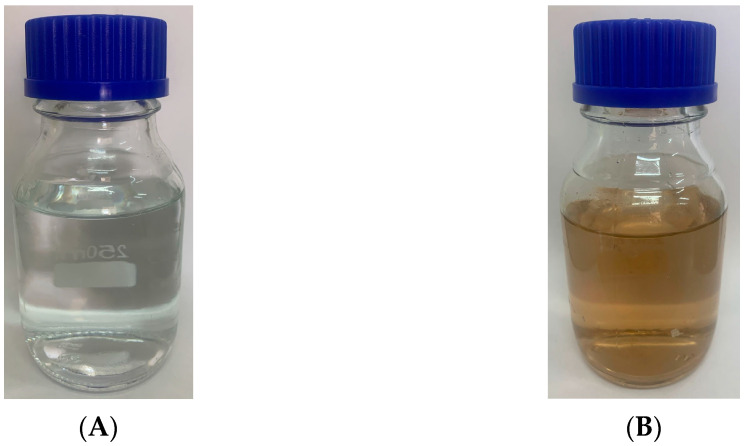
Exhausted culture media after the production of culture media *L. rhamnosus* (**A**); AgNPs synthesized with the exhausted culture media (**B**).

**Figure 2 ijms-25-11809-f002:**
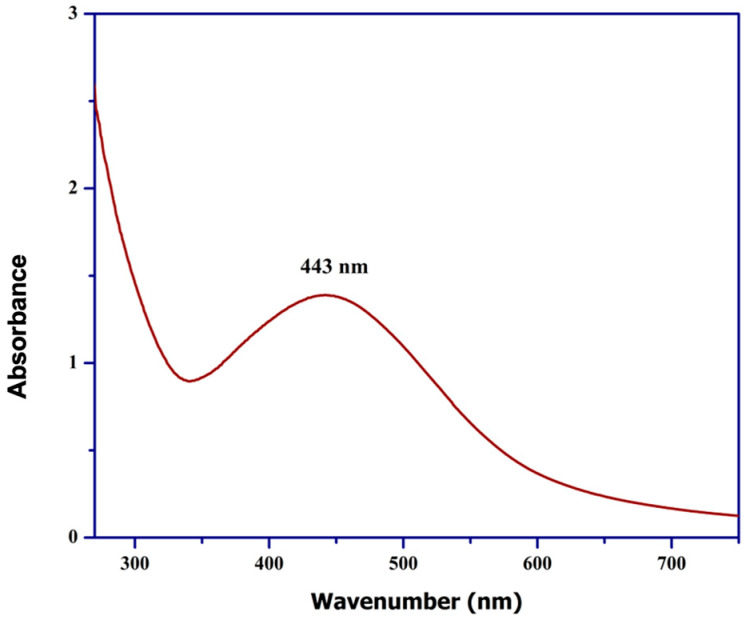
AgNP UV–visible absorption peak at 443 nm.

**Figure 3 ijms-25-11809-f003:**
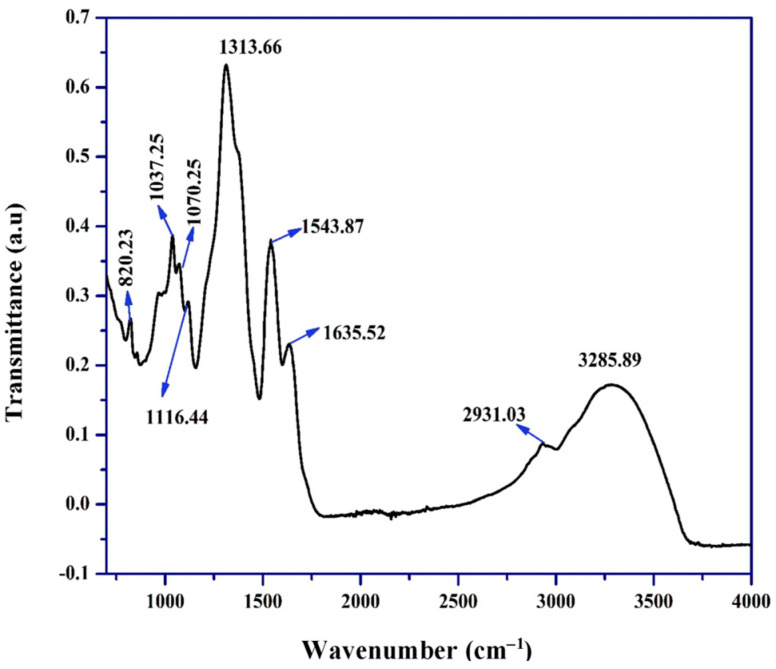
FTIR spectrum of biosynthesized silver nanoparticles (AgNPs). The FTIR spectrum shows prominent peaks corresponding to functional groups including O–H, C–H, C=O, and S=O.

**Figure 4 ijms-25-11809-f004:**
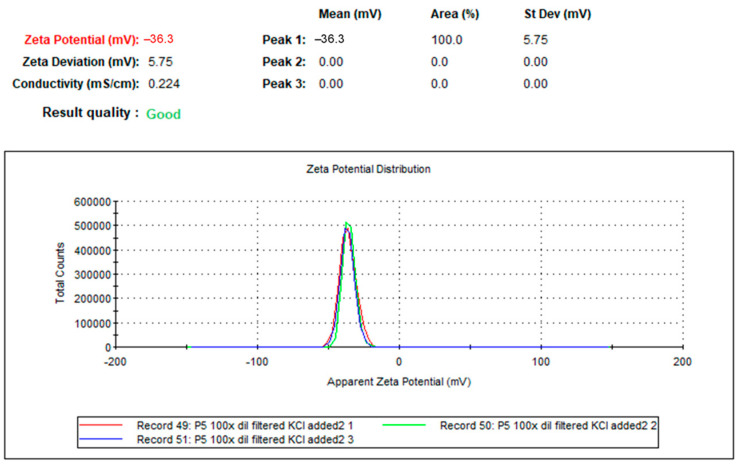
Zeta potential analysis of biosynthesized silver nanoparticles (AgNPs). The zeta potential of −36.3 mV indicates good stability and dispersion of the AgNPs in solution.

**Figure 5 ijms-25-11809-f005:**
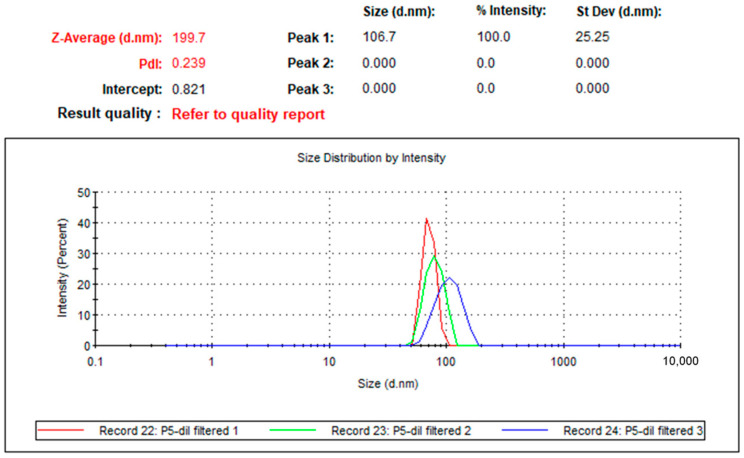
Dynamic light scattering (DLS) analysis of biosynthesized silver nanoparticles (AgNPs). The dynamic light scattering (DLS) analysis indicates a mean particle size of 199.7 nm with a polydispersity index (PDI) of 0.239, demonstrating the practical synthesis and stability of AgNPs using the probiotic-mediated approach.

**Figure 6 ijms-25-11809-f006:**
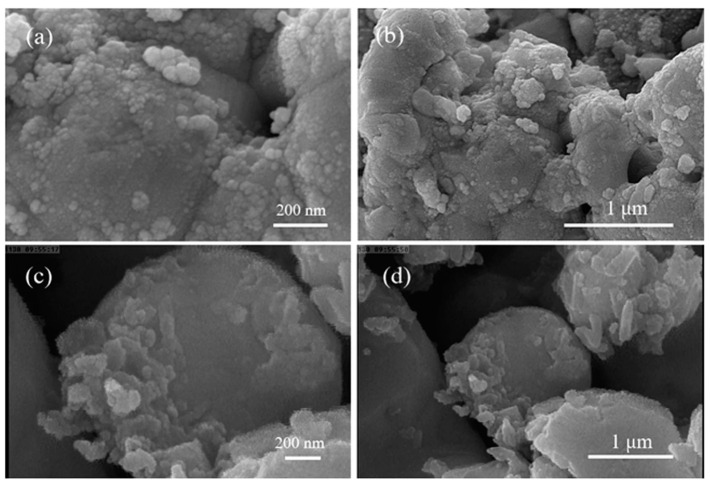
SEM images depicting morphology and surface structure of synthesized AgNPs at higher (**a**,**c**) and lower (**b**,**d**) magnification.

**Figure 7 ijms-25-11809-f007:**
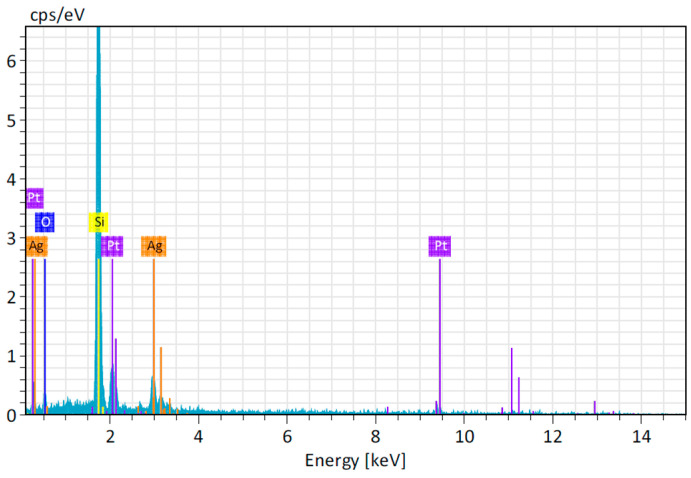
EDS spectrum of silver nanoparticles.

**Figure 8 ijms-25-11809-f008:**
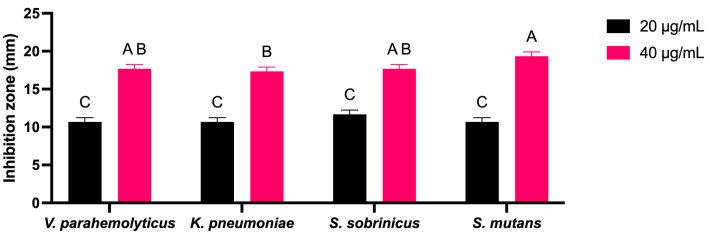
Two-way ANOVA for the zone of inhibition measurements (in mm) for various microorganisms treated with two volumes (20 µL and 40 µL) of AgNPs. Columns with letters (A, B, C) are statistically not significant between each other. Significant differences were found between A and C.

**Figure 9 ijms-25-11809-f009:**
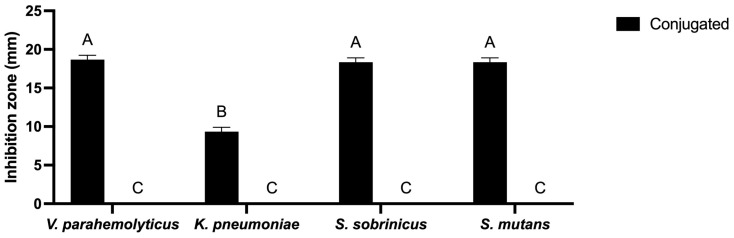
Two-way ANOVA for the zone of inhibition measurements (in mm) for various microorganisms treated on cotton fabrics. Columns with letters (A, B, C) are statistically not significant between each other. Significant differences were found between A–B, A–C, and B–C.

**Figure 10 ijms-25-11809-f010:**
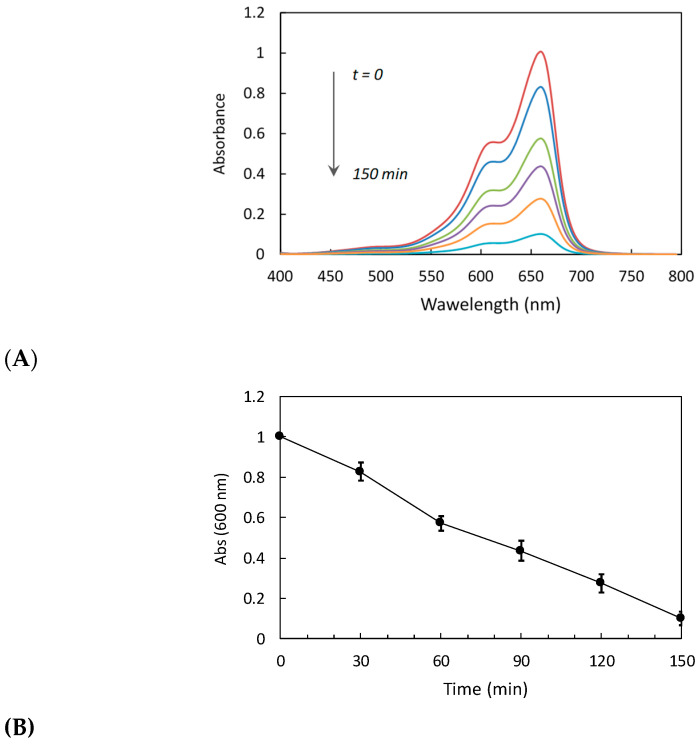
Photocatalytic degradation of methylene blue dye using *L. rhamnosus* (BCRC16000)-mediated silver nanoparticles under visible light at different time intervals (t = 0, 30, 60, 90, 120, and 150 min). Changes in the absorption spectra of methylene blue in the visible region (**A**) and the absorbance values at 600 nm (**B**).

## Data Availability

The original contributions presented in this study are included in the article; further inquiries can be directed to the corresponding author.

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
