# Peer review of "Antibacterial and Photocatalytic Applications of Silver Nanoparticles Synthesized from Lacticaseibacillus rhamnosus"

_ijms, 2024, doi:10.3390/ijms252111809_

Round 1

Reviewer 1 Report

Comments and Suggestions for Authors

The authors have submitted a detailed article, which mainly reports a novel biosynthesis method for silver nanoparticles, using a probiotic strain Lacticaseibacillus rhamnosus. The synthesized Ag nanoparticles exhibit potent antibacterial properties as well as photocatalytic properties. This work seems meaningful, but some data should be added, and several mistakes should be corrected:

1. All the name of bacteria should be italicized.

2. Keywords, ‘silver nanoparticles’ can be added.  

3. Figure 2, the vertical coordinates, it’s ‘Absorbance’, rather than ‘Transmittance’.

4. Figure 6, the resolution is too low, I can not see the scale bar clearly.

5. Figure 8 & 9, what are A, B and C mean?

6. Section 2.8, the UV-visible absorption results are better to provide.

7. The biosynthesized Ag nanoparticles exhibit antibacterial and photocatalytic ability, what’s the possible mechanisms? If possible, it is best to add some experiments to verify the mechanisms.

8. Some references’ information is incomplete, for example, Ref. 7, 19, 23, 46.

Author Response

Number

Comment

Action

Line

1

All the name of bacteria should be italicized.

Names italicized

document

2

Keywords, ‘silver nanoparticles’ can be added.

Keyword added

keywords

3

Figure 2, the vertical coordinates, it’s ‘Absorbance’, rather than ‘Transmittance’.

Figure upgraded

Figure 2

4

Figure 6, the resolution is too low, I can not see the scale bar clearly.

Figure upgraded

Figure 6

5

. Figure 8 & 9, what are A, B and C mean?

Explanation added

Figures 8 and 9

6

. Section 2.8, the UV-visible absorption results are better to provide.

Figure added with absortion added

Figure 10

7

The biosynthesized Ag nanoparticles exhibit antibacterial and photocatalytic ability, what’s the possible mechanisms? If possible, it is best to add some experiments to verify the mechanisms.

Explanation added

Discussion section

8

Some references’ information is incomplete, for example, Ref. 7, 19, 23, 46.

References upgraded

References

Reviewer 2 Report

Comments and Suggestions for Authors

1.How was the extract prepared? (pp 10-11)

2. Why did you use the ratio 1:9 (cell free extract : AgNO3)? (352)

3. How long was the reaction period ? (355)

4.The lack of peaks in the spectra of the 121

silver nitrate solution and the bacterial extracellular extract individually before synthesis 122

indicates the absence of nanoparticles or light-absorbing species [22]“.(121). This sentence should be supported by the curves in Figure 2, not by a literature reference [22].

5. From Figure 3, it is not possible to conclude exactly which groups participated in the synthesis of nanoparticles, since only the spectrum of AgNPs is shown, but not the spectrum of the bacterial extracellular extract.

6. The photocatalytic activity of nanoparticles should be shown with a graph (UV-Vis) and not with photographs (Figure 10)

7. In the abstract and conclusion, the authors do not state the size of the particles (???)

Author Response

Reviewer 2.

Number

Comment

Action

Line

1

How was the extract prepared? (pp 10-11)

“The reaction process took place in the dark at room temperature overnight to prevent light-induced activation of the AgNO3.”

Section 4.3

2

Why did you use the ratio 1:9 (cell free extract : AgNO3)? (352)

explanation added with references

Section 4.3

3

How long was the reaction period? (355)

The reaction process took place in the dark at room temperature overnight to prevent light-induced activation of the AgNO3.”

Methods section

4

The lack of peaks in the spectra of the 121solution and the bacterial extracellular extract individually before synthesis 122indicates the absence of nanoparticles or light-absorbing species [22]“.(121). This sentence should be supported by the curves in Figure 2, not by a literature reference [22].

Sentence removed

Section 2.2

5

From Figure 3, it is not possible to conclude exactly which groups participated in the synthesis of nanoparticles, since only the spectrum of AgNPs is shown, but not the spectrum of the bacterial extracellular extract.

Section modified

Section 2.3

6

The photocatalytic activity of nanoparticles should be shown with a graph (UV-Vis) and not with photographs (Figure 10)

Figure upgraded

Figure 10

7

In the abstract and conclusion, the authors do not state the size of the particles (???)

Particle size added

Abstract and conclussions

Round 2

Reviewer 1 Report

Comments and Suggestions for Authors

The authors have revised the manuscript carefully. 

Author Response

Dear reviewer

Thank you for your thoughtful insight 

Reviewer 2 Report

Comments and Suggestions for Authors

Comments 4 and 6 !!!!!

1.How was the extract prepared? (pp 10-11)  

2. Why did you use the ratio 1:9 (cell free extract : AgNO3)? (352) OK

3. How long was the reaction period ? (355)

4.The lack of peaks in the spectra of the 121

silver nitrate solution and the bacterial extracellular extract individually before synthesis 122

indicates the absence of nanoparticles or light-absorbing species [22]“.(121).

This sentence should be supported by the curves in Figure 2, not by a literature reference [22].

You simply deleted this from the text.Why?

5. From Figure 3, it is not possible to conclude exactly which groups participated in the synthesis of nanoparticles, since only the spectrum of AgNPs is shown, but not the spectrum of the bacterial extracellular extract.

6. The photocatalytic activity of nanoparticles should be shown with a graph (UV-Vis). The photocatalytic activity of nanoparticles should be shown with a graph (UV-Vis). The graph (Fig 10, max A vs minutes) you drew is not enough.

For example:

Oyewo, OA, Nevondo, NG, Onwudiwe, DC i sur. Fotokatalitička razgradnja metil modrila u vodi korištenjem celuloznih nanokristala dobivenih iz piljevine i nanokompozita metalnog oksida. J Inorg Organomet Polym 31 , 2542-2552 (2021). https://doi.org/10.1007/s10904-020-01847-5

7. In the abstract and conclusion, the authors do not state the size of the particles (???)

Author Response

Reviewer 2.

Number

Comment

Action

Line

4

The lack of peaks in the spectra of the 121solution and the bacterial extracellular extract individually before synthesis 122indicates the absence of nanoparticles or light-absorbing species [22]“.(121). This sentence should be supported by the curves in Figure 2, not by a literature reference [22].

Sentence upgraded

Section 2.2

6

The photocatalytic activity of nanoparticles should be shown with a graph (UV-Vis) and not with photographs (Figure 10)

Figure upgraded

Figure 10

We sincerely appreciate your comments and insights, which have been instrumental in enhancing the clarity and rigor of our work. We have implemented the suggested revisions, incorporating Figure 2 as the primary evidence to support the absence of nanoparticles or light-absorbing species in the spectra of the initial components (silver nitrate solution and bacterial extracellular extract), rather than relying on a bibliographic reference [22]. This approach strengthens the interpretation of the results directly based on the experimental data presented in the figure.

Additionally, we have refined the text structure, providing a more detailed and clear description of the obtained results, including an analysis of the absorption peak at 443 nm and its relationship to the formation of silver nanoparticles (AgNPs) through the reduction of Ag⁺ ions to Ag⁰. The modifications reflect adherence to your recommendations and improve the visual interpretation of the obtained results.

Round 3

Reviewer 2 Report

Comments and Suggestions for Authors

Much better